# Predicting in-Hospital Mortality of Patients with COVID-19 Using Machine Learning Techniques

**DOI:** 10.3390/jpm11050343

**Published:** 2021-04-24

**Authors:** Fabiana Tezza, Giulia Lorenzoni, Danila Azzolina, Sofia Barbar, Lucia Anna Carmela Leone, Dario Gregori

**Affiliations:** 1Geriatric Unit, Ospedali Riuniti di Padova Sud, AULSS 6 Euganea, 35043 Monselice, Italy; fabiana.tezza@aulss6.veneto.it; 2Unit of Biostatistics, Epidemiology and Public Health, Department of Cardiac, Thoracic, Vascular Sciences and Public Health, University of Padova, 35131 Padova, Italy; giulia.lorenzoni@unipd.it (G.L.); danila.azzolina@uniupo.it (D.A.); 3Department of Translational Medicine, University of Eastern Piedmont, 28100 Novara, Italy; 4Internal Medicine Unit, Cittadella Hospital, AULSS 6 Euganea, 35013 Cittadella, Italy; sofia.barbar@aulss6.veneto.it; 5Internal Medicine Unit, Ospedali Riuniti Padova Sud, AULSS 6 Euganea, 35043 Monselice, Italy; lucia.leone@aulss6.veneto.it

**Keywords:** machine learning techniques, COVID-19, Italy, in-hospital mortality, outcome prediction

## Abstract

The present work aims to identify the predictors of COVID-19 in-hospital mortality testing a set of Machine Learning Techniques (MLTs), comparing their ability to predict the outcome of interest. The model with the best performance will be used to identify in-hospital mortality predictors and to build an in-hospital mortality prediction tool. The study involved patients with COVID-19, proved by PCR test, admitted to the “Ospedali Riuniti Padova Sud” COVID-19 referral center in the Veneto region, Italy. The algorithms considered were the Recursive Partition Tree (RPART), the Support Vector Machine (SVM), the Gradient Boosting Machine (GBM), and Random Forest. The resampled performances were reported for each MLT, considering the sensitivity, specificity, and the Receiving Operative Characteristic (ROC) curve measures. The study enrolled 341 patients. The median age was 74 years, and the male gender was the most prevalent. The Random Forest algorithm outperformed the other MLTs in predicting in-hospital mortality, with a ROC of 0.84 (95% C.I. 0.78–0.9). Age, together with vital signs (oxygen saturation and the quick SOFA) and lab parameters (creatinine, AST, lymphocytes, platelets, and hemoglobin), were found to be the strongest predictors of in-hospital mortality. The present work provides insights for the prediction of in-hospital mortality of COVID-19 patients using a machine-learning algorithm.

## 1. Introduction

The prognosis prediction of patients infected with SARS-CoV-2 is one of the most relevant topics in COVID-19 research [1]. The pandemic poses a severe burden to the healthcare system of countries worldwide given the long symptoms’ duration [2], the higher risk of severe complications requiring hospitalization/intensive care compared to seasonal influenza [3], and the documented excess of mortality associated with the virus spread [4]. Such conditions have led to a dramatic imbalance between healthcare resources and hospital/Intensive Care Unit (ICU) bed demands [5]. Italy, the first and most affected European country during the first COVID-19 wave [6], adopted emergency solutions to face such imbalance [5]. However, the COVID-19 wave is still ongoing in almost all European countries, including Italy. For these reasons, there is an urgent need for strategies to prevent such an imbalance occurs again. With the adoption of public health strategies to contain the virus spread [7] and the improvement of the health care resources available, the development of tools to predict patients’ prognosis would be part of the solution to prevent resource imbalance [8]. Such tools would be helpful to assist physicians’ decision-making. Risk stratification is essential in this context to identify an adequate referral pathway for each COVID-19 patient, allowing for an appropriate resource allocation. Furthermore, such tools help the identification of the parameters that contribute the most to the outcome, allowing physicians to monitor/treat such parameters.

Several tools have been proposed from the beginning of the pandemic, especially for in-hospital mortality prediction [9]. A systematic review published at the beginning of the pandemic, i.e., April 2020, identified several mortality risk prediction tools for COVID-19 patients [9], and several others have been proposed after that. Several methods have been employed to develop such tools, from traditional statistical approaches, e.g., multivariable analysis [10], to more advanced machine learning techniques (MLTs) [11,12,13,14]. MLTs are increasingly used for outcome prediction in the clinical setting, since they present several advantages over traditional methods [15]. They help disentangle complex relationships between covariates and outcome of interest, even though a low number of events have occurred in front of many variables to be tested. Not least, the MLT predictive ability may improve as new data is provided.

Our study aims to identify predictors of COVID-19 in-hospital mortality testing a set of MLTs, comparing their ability to predict the outcome of interest. The model estimation is based on patients treated in a COVID-19 referral center in the Veneto region, Italy, during the COVID-19 first wave. Together with Lombardia, Veneto was the first Italian region affected by the COVID-19 outbreak at the beginning of the year [16].

## 2. Materials and Methods

The study involved patients with COVID-19, proved by PCR test, admitted to the “Ospedali Riuniti Padova Sud” COVID-19 referral center in the Veneto region, Italy, in March and April 2020.

### 2.1. Data Collection

Information on sociodemographic characteristics; medical history (comorbidities, smoking habits, and weight status); clinical and instrumental parameters; and laboratory data were collected at hospital admission. Furthermore, drug therapy was registered. The outcome was represented by in-hospital mortality.

### 2.2. Statistical Analysis

Descriptive statistics were reported as I quartile/median/III quartile for continuous variables and percentages (absolute numbers) for categorical variables. Wilcoxon-type tests were performed for the continuous variables and the Pearson chi-square test, or Fisher-exact test, whichever is appropriate, for the categorical variables.

A set of MLTs was tuned to identify the predictors of in-hospital mortality. The predictive tools were tuned using a bootstrapping algorithm (50 runs). The algorithms considered for the classification task were the Recursive Partition Tree (RPART), the Support Vector Machine (SVM), the Gradient Boosting Machine (GBM), and Random Forest.

The RPARTs are classification models based on a top-down methodology in which, starting from a root node, binary splits of data are generated until a certain criterion (i.e., the minimization of the node impurity) is encountered [17]. This method is prone to overfitting on training data. The cross-validation or bootstrapping procedure is a useful method to limit the overfitting, leading to defining the proper tuning of the Decision Tree (DT) parameters and optimizing the model accuracy [18].Random Forest is a tree-based algorithm, which involves the computation of hundreds to thousands of RPART trees; the method merges the DT output to increase the model’s generalizability [19].The GBM is based on a sequential boosting improvement of weak RPART classifiers (high bias and low variance). The GBM idea is to add a classifier sequentially so that the next classifier is trained to improve the already trained RPART. A random forest algorithm, instead, trains each classifier independently from the others [19].The SVM algorithm’s main objective is to find an optimal hyperplane of the feature’s *N*-dimensional space (*N*—the number of variables) that distinctly classify the data points into a binary partition [20]. Several hyperplanes may separate the classes of data points. The SVM algorithm considers the hyperplanes, which maximize the margin (the distance between data points of classes).

The resampled performances were reported for each MLT, considering the sensitivity, specificity, and the Receiving Operative Characteristic curve (ROC) measures, with 95% confidence intervals (C.I.s).

Among the proposed MLTs, the tool with the best performance was chosen for the classification task. c, pairwise differences were computed and tested to assess if the difference was equal to zero. The Bonferroni correction was used to calculate the *p*-values and adjust the confidence interval limits for the reported differences. For the most promising MLT, the variable importance plot was reported together with the ROC curve and the median balanced accuracy measure within the resampling. 

The ROC and accuracy measures were also reported to evaluate the MLT tool’s performance according to the increasing data availability during the study period. The database was ordered according to the hospitalization date. The MLT was built on the first 40, 60, 80,100, 120, 140, 160, 180, 200, 220, 240, 260, 280, 300, 320, and 340 hospitalized patients. The ROC and accuracy measures were reported according to the different sample sizes. The performance was computed, together with the variable importance measures, by tuning and developing the tool for cumulative fractions of the sample.

A Shiny web application was developed based on the algorithm with the best performance. The tool calculates the in-hospital death probability, according to the patients’ characteristics.

Analyses were performed using R 3.4.2 [21] with the rms [22] and caret [23] packages.

## 3. Results

The study enrolled 341 patients admitted to the “Ospedali Riuniti Padova Sud” COVID-19 referral center between March and April 2020. Table 1 presents the subjects’ baseline characteristics according to the admission date (before–after 21 March 2020, the day on which the national lockdown was declared in Italy). Seventy-five patients out of 341 died while hospitalized.

Overall, the prevalence of the male gender was found to be higher than the female one (57% in the sample overall), even though it was found to be higher in the first period of admissions compared to the second one (65% vs. 51%; *p*-value 0.007). The median age of the subjects enrolled was 74 years, and it was lower in the first period compared to the second period (median of 70 vs. 78; *p*-value < 0.001). Subjects admitted to the hospital in the first period were significantly more likely to suffer from diabetes (*p*-value 0.031) and less likely to suffer from dementia (*p*-value < 0.001) and from cerebrovascular diseases (*p*-value 0.019). No significant differences were detected in the distribution of the other comorbidities. For what concerns the drug therapy, subjects admitted after March 21 were found to be significantly more likely to be treated with low molecular weight heparin (LMWH) (*p*-value 0.001) but less likely to be administered with antiviral drugs.

### 3.1. MLTs Performance

The Random Forest algorithm outperformed the other MLTs in predicting the in-hospital mortality (Figure 1), with a ROC of 0.84 (95% C.I. 0.78–0.9). In addition to that, looking at the values of sensitivity and specificity, Random Forest provided a better balance than the other techniques between two such measures. Such results were confirmed when the pairwise comparisons were made (Appendix A). Furthermore, no class imbalance issues were detected (Appendix A).

### 3.2. Variable Importance in Predicting in-Hospital Mortality According to the Random Forest

Since Random Forest was found to be the MLT with the best performance, it was chosen to identify the predictors of in-hospital mortality. The algorithm was tuned to achieve the optimal performance in the correspondence of 500 trees and 6 mtry (number of variables available for splitting each tree node).

Figure 2 reports the plots of the variable importance measures according to the mean decrease accuracy. Age, together with vital signs (oxygen saturation and the quick SOFA) and lab parameters (creatinine, AST, lymphocytes, platelets, and hemoglobin), were found to be the strongest predictors of in-hospital mortality (Figure 2). Conversely, comorbidities were found to provide only a small contribution in predicting the in-hospital mortality in such patients.

Furthermore, the algorithm stability investigation as the availability of data increases was performed to inform clinicians about the minimum number of cases required to obtain a reliable algorithm. Interestingly, the ROC parameter underwent stabilization after the first 100 patients (Figure 3). The variable importance plots calculated on cumulative fractions of the sample are provided in Appendix A.

The web application based on the algorithm developed is available at https://r-ubesp.dctv.unipd.it/shiny/Schiavonia/.

## 4. Discussion

The present study showed that the Random Forest is a feasible machine-learning algorithm to stratify the mortality risk in a sample of patients admitted to a COVID-19 referral center in the Veneto region, Italy, during the first COVID-19 wave.

The sample characteristics, in terms of gender and comorbidities distribution, are consistent with those reported by previous studies on subjects admitted to the Italian hospitals during the first wave of the pandemic [24], even though the median age of the patients enrolled was about ten years higher than that previously reported [24]. For what concerns drug therapy, the increasing use of LMWH documented between the first and second study periods reflects the publication of the preliminary evidence about the beneficial effects of LMWH in patients with COVID-19 [25].

Risk stratification is a hot topic in COVID-19 research, given the shortage of healthcare resources for the management of COVID-19 patients. For this reason, an impressive number of risk stratification tools for in-hospital mortality in COVID-19 patients has been developed. Both traditional, i.e., logistic regression, and nontraditional, i.e., MLTs, statistical techniques have been employed to create such tools [9], but the traditional ones have been most frequently used, probably for the ease of use and interpretability in the clinical setting. However, MLTs have been shown to be feasible to develop risk stratification tools in the COVID-19 research setting [11,13], with promising results underlying such algorithms’ potentials to assist clinicians’ decision-making in everyday clinical practice.

The present work showed an outperformance of Random Forest compared to the other algorithms tested. Interestingly, the variables found to contribute to in-hospital mortality were age and those related to hospital presentation (vital signs and lab parameters). At the same time, the comorbidities played a minor role in the mortality risk prediction. Such findings are only partially in-line with those of international studies, showing that, together with clinical presentation, comorbidities also play a relevant role in risk prediction [11]. However, it is worth pointing out that it is difficult to compare results from the literature, since each tool has been developed on different patient populations, e.g., hospitalized patients, mechanically ventilated patients, elderly patients hospitalized, and the general population. Furthermore, each tool has been developed using a different set of baseline variables, including sociodemographic ones, laboratory data, instrumental parameters, and clinical measures.

## 5. Conclusions

The present work provides a useful tool able to assist physicians’ decision-making in facing the COVID-19 emergency. Furthermore, the feasibility of MLTs in this research context has been shown, which represents an added value of the present work, since MLTs are able to overcome the limitations of more traditional statistical techniques.

## Figures and Tables

**Figure 1 jpm-11-00343-f001:**
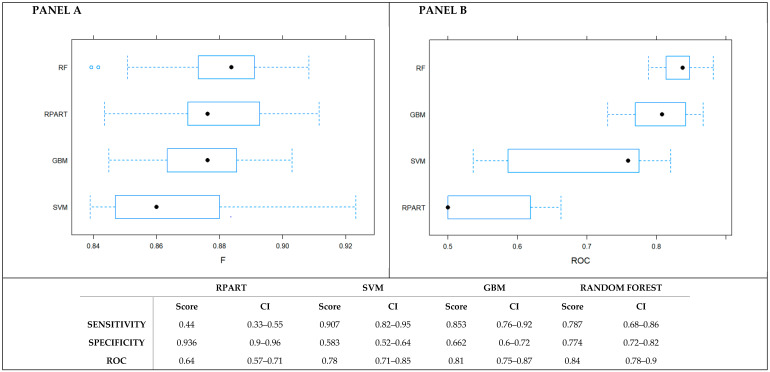
Comparison of the performance of the MLT algorithms. **Panel A** presents the F-score measure. **Panel B** presents the ROC.

**Figure 2 jpm-11-00343-f002:**
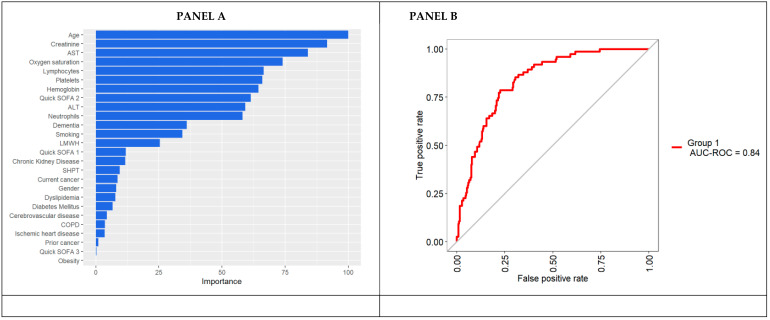
Random Forest’s variable importance plot (**Panel A**) and ROC curve on the Random Forest predictions (**Panel B**). The median resampled accuracy is 0.86.

**Figure 3 jpm-11-00343-f003:**
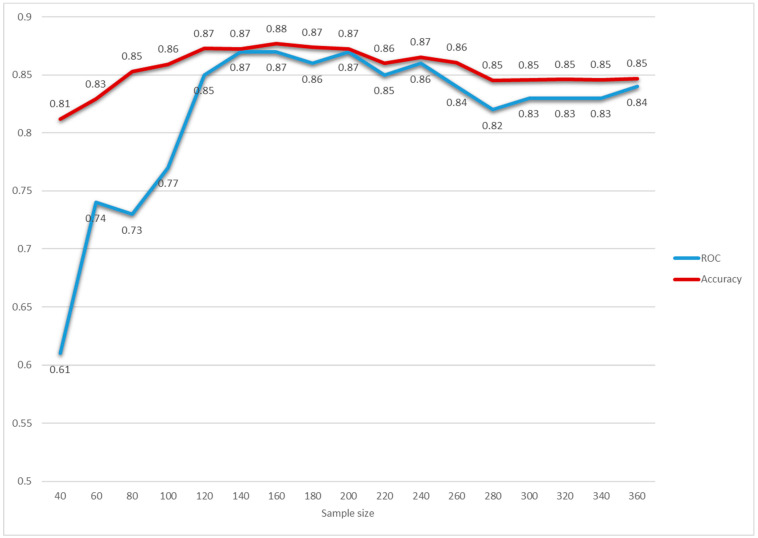
Random Forest’s ROC and accuracy as the availability of the data increased. The database was ordered according to the hospitalization date. Random Forests were built on the first 40, 60, 80,100, 120, 140, 160, 180,200, 220, 240, 260, 280, 300, 320, and 340 hospitalized patients. The ROC (blue line) and accuracy (red line) measures were reported according to the different sample sizes. The performance was computed by tuning and developing the Random Forest tool for the cumulative fractions of the sample.

**Table 1 jpm-11-00343-t001:** Subjects’ baseline characteristics according to the admission date (before–after 21 March 2020). Data are I quartile/Median/III quartile for continuous variables and percentages (absolute numbers) for categorical variables.

Baseline Characteristics	*N*	Before March 21 (*n* = 149)	After March 21 (*n* = 192)	Overall (N = 341)	*p*-Value
Gender: Female	341	35% (52)	49% (95)	43% (147)	0.007
Male		65% (97)	51% (97)	57% (194)	
Age	341	58.0/70.0/82.0	62.5/78.0/87.0	60.0/74.0/85.0	<0.001
Smoking: No	237	79% (94)	85% (100)	82% (194)	0.25
Yes		21% (25)	15% (18)	18% (43)	
Duration of Hospitalization (Days)	290	7.0/12.0/17.0	6.0/10.0/16.0	7.0/11.0/16.8	0.106
Comorbidities					
Diabetes Mellitus: No	341	73% (109)	83% (159)	79% (268)	0.031
Yes		27% (40)	17% (33)	21% (73)	
SHPT: No	341	48% (72)	53% (102)	51% (174)	0.379
Yes		52% (77)	47% (90)	49% (167)	
Dyslipidemia: No	341	82% (122)	86% (165)	84% (287)	0.309
Yes		18% (27)	14% (27)	16% (54)	
Ischemic heart disease: No	341	91% (136)	90% (172)	90% (308)	0.6
Yes		9% (13)	10% (20)	10% (33)	
Obesity: No	341	93% (138)	94% (181)	94% (319)	0.538
Yes		7% (11)	6% (11)	6% (22)	
Cerebrovascular disease: No	341	95% (142)	88% (169)	91% (311)	0.019
Yes		5% (7)	12% (23)	9% (30)	
Dementia: No	341	87% (129)	69% (132)	77% (261)	<0.001
Yes		13% (20)	31% (60)	23% (80)	
Chronic Kidney Disease: No	331	93% (136)	89% (164)	91% (300)	0.293
Yes		7% (11)	11% (20)	9% (31)	
Current cancer: No	341	96% (143)	91% (175)	93% (318)	0.078
Yes		4% (6)	9% (17)	7% (23)	
Previous Cancer: No	340	95% (142)	95% (182)	95% (324)	0.995
Yes		5% (7)	5% (9)	5% (16)	
COPD: No	340	95% (140)	90% (172)	92% (312)	0.096
Yes		5% (8)	10% (20)	8% (28)	
Lab parameters & Vital Signs					
Neutrophils (×10^9^/L)	339	3.16/4.38/6.56	3.24/4.57/7.23	3.17/4.50/6.72	0.235
Lymphocytes (×10^9^/L)	339	0.640/0.915/1.263	0.670/0.890/1.345	0.650/0.900/1.320	0.784
Neutrophils/Lymphocytes (×10^9^/L)	338	2.69/4.42/7.96	3.23/5.09/9.36	2.97/4.93/8.93	0.297
Hemoglobin (g/L)	339	116/130/142	116/128/140	116/128/140	0.586
D-dimer (μg/L)	124	134/192/416	148/301/675	142/227/636	0.089
Creatinine (μmol/L)	337	67.8/84.0/104.2	63.0/78.0/96.0	67.0/81.0/102.0	0.075
Platelets (×10^9^/L)	338	148/189/234	150/198/246	148/196/244	0.445
AST (U/L)	292	25.2/35.5/49.0	26.0/37.0/60.0	26.0/36.5/55.2	0.265
ALT (U/L)	298	20.0/27.0/40.0	15.8/25.0/43.2	17.0/26.0/42.0	0.2
Procalcitonin (PCT) (μg/L)	208	0.0400/0.0800/0.1400	0.0500/0.1200/0.3950	0.0475/0.1000/0.2600	0.03
Troponine (ng/L)	31	6.0/25.0/90.0	10.5/26.5/64.5	9.5/25.0/68.0	0.848
Oxygen saturation (%)	244	93.8/96.0/98.0	93.0/95.0/97.0	93.0/95.5/98.0	0.541
Quick SOFA	341	0/0/1	0/0/1	0/0/1	0.048
pO2 (mmHg)	213	54.0/64.0/71.0	56.0/64.0/73.2	55.0/64.0/72.0	0.567
Glasgow Coma Scale >/= 15 : No	341	84% (125)	70% (135)	76% (260)	0.003
Yes		16% (24)	30% (57)	24% (81)	
Drug therapy					
ARIXTRA: No	111	59% (37)	35% (17)	49% (54)	0.015
Yes		41% (26)	65% (31)	51% (57)	
LMWH: No	286	28% (35)	12% (20)	19% (55)	0.001
Yes		72% (88)	88% (143)	81% (231)	
Antiviral: No	340	51% (76)	97% (186)	77% (262)	<0.001
Yes		49% (73)	3% (5)	23% (78)	
Plaquenil: No	340	24% (36)	18% (34)	21% (70)	0.15
Yes		76% (113)	82% (157)	79% (270)	
Antibiotic: No	340	5% (7)	4% (8)	4% (15)	0.82
Yes		95% (142)	96% (183)	96% (325)	
Tocilizumab: No	340	95% (141)	97% (186)	96% (327)	0.189
Yes		5% (8)	3% (5)	4% (13)	

## Data Availability

The data presented in this study are available upon request from the corresponding author.

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
