# Peer review of "Predicting in-Hospital Mortality of Patients with COVID-19 Using Machine Learning Techniques"

_jpm, 2021, doi:10.3390/jpm11050343_

Round 1

Reviewer 1 Report

Interesting study on "hot topic" matters for public health system: the variables that better predict in hospital covid related deaths. The authors explain correctly which  methodology appears, in their evidences, to fit more appropriate for their purpose and in their community.

The present study tried with machine learning techniques to identify predictors of covid related pneumonia in-hospital mortality. Several models of algorithms were applied on a sample of 341 patients recovered in Padua Hospital, Italy. The authors present an extensive analisys of MLT considering sensitivity, specifity and "curves" measures. The pourpose was correctly explain, as well as methodology. The results provide an interesting and significant tool to use in clinical practice, i.e. Random Forest, and also new for covid disease . The paper is well written, clear and the conclusions are consistent with the main object of the study

Author Response

Reviewer 1

Open Review

English language and style

( ) Extensive editing of English language and style required
( ) Moderate English changes required
(x) English language and style are fine/minor spell check required
( ) I don't feel qualified to judge about the English language and style

Yes

Can be improved

Must be improved

Not applicable

Does the introduction provide sufficient background and include all relevant references?

(x)

( )

( )

( )

Is the research design appropriate?

(x)

( )

( )

( )

Are the methods adequately described?

(x)

( )

( )

( )

Are the results clearly presented?

(x)

( )

( )

( )

Are the conclusions supported by the results?

(x)

( )

( )

( )

Comments and Suggestions for Authors

Interesting study on "hot topic" matters for public health system: the variables that better predict in hospital covid related deaths. The authors explain correctly which  methodology appears, in their evidences, to fit more appropriate for their purpose and in their community.

The present study tried with machine learning techniques to identify predictors of covid related pneumonia in-hospital mortality. Several models of algorithms were applied on a sample of 341 patients recovered in Padua Hospital, Italy. The authors present an extensive analisys of MLT considering sensitivity, specifity and "curves" measures. The pourpose was correctly explain, as well as methodology. The results provide an interesting and significant tool to use in clinical practice, i.e. RF, and also new for covid disease . The paper is well written, clear and the conclusions are consistent with the main object of the study

We would like to thank the reviewer for his/her comments.

Reviewer 2 Report

Tezza and colleagues report original research results on machine learning models to predict in-hospital mortality of COVID-19 patients. They analyzed a sample of 341 patients (median age 74 years) and compared different algorithms, to conclude that a random forest classifier outperformed RPART and SVM models. The manuscript is well written and the authors clearly explain the main aspects of their methods, results and findings. However, some points can be improved: 

  • The authors could try to improve the description of key demographics. For instance, it is not easy for the readers to determine, out of the 341 patients, how many subjects died and how many of them did not. 
  • Was there class imbalance? How would that affect the models? 
    The table presented as part of Figure 1 reports confidence intervals for sensitivity, specificity and ROC for RPART, SVM, GBM and RF models. Some of those confidence intervals show substantial overlap (e.g., ROC intervals are very similar for SVM, GBM and RF).
  • The authors may want to conduct a statistical test on the classifier performance metrics to validate whether or not their outcomes are significantly different. Several literature references on the topic of model comparison can inform on how to conduct this step (e.g., https://www.jmlr.org/papers/volume7/demsar06a/demsar06a.pdf ).

Author Response

Reviewer 2

Open Review

English language and style

( ) Extensive editing of English language and style required
( ) Moderate English changes required
(x) English language and style are fine/minor spell check required
( ) I don't feel qualified to judge about the English language and style

Yes

Can be improved

Must be improved

Not applicable

Does the introduction provide sufficient background and include all relevant references?

(x)

( )

( )

( )

Is the research design appropriate?

(x)

( )

( )

( )

Are the methods adequately described?

( )

( )

(x)

( )

Are the results clearly presented?

( )

(x)

( )

( )

Are the conclusions supported by the results?

( )

( )

(x)

( )

Comments and Suggestions for Authors

Tezza and colleagues report original research results on machine learning models to predict in-hospital mortality of COVID-19 patients. They analyzed a sample of 341 patients (median age 74 years) and compared different algorithms, to conclude that a RF classifier outperformed RPART and SVM models. The manuscript is well written and the authors clearly explain the main aspects of their methods, results and findings. However, some points can be improved: 

  • The authors could try to improve the description of key demographics. For instance, it is not easy for the readers to determine, out of the 341 patients, how many subjects died and how many of them did not. 

The manuscript has been amended according to reviewer’s comment, also reporting the number of patients who underwent in-hospital death.

The study enrolled 341 patients admitted to the “Ospedali Riuniti Padova Sud” COVID-19 referral center between March and April 2020. Table 1 presents subjects’ baseline characteristics according to the admission date (before-after March 21, 2020, the day on which the national lockdown was declared in Italy). Seventy-five patients out of 341 died while hospitalized. Overall, the prevalence of the male gender was found to be higher than the female one (57% in the sample overall), even though it was found to be higher in the first period of admissions compared to the second one (65% vs. 51%, p-value 0.007). The median age of the subjects enrolled was 74 years, and it was lower in the first period compared to the second period (median of 70 vs. 78, p-value <0.001).”

  • Was there class imbalance? How would that affect the models? 

There was no evidence of class imbalance that could affect the performance of the model. For most MLT algorithms, literature suggests that the loss of model performance due to imbalance begins to be significant when the class with the smaller number of observations represents 10% of the data or less (1). Conversely, in our data, the minority class represents 22% of the entire sample. However, we found to be useful to run a sensitivity analysis on the Random Forest (RF) performance, which represents the most performing algorithm used to develop the final predictive tool.

The RF algorithm has been recomputed by considering a model weight procedure to handle class imbalance (2). The performances for the RF model adjusted for class imbalance model weight procedure are reported below, showing results that are consistent to the original model presented in the manuscript.

RF’s ROC curve on predictions for weight model adjusted RF

AUC=0.843

·         median resampled accuracy: 86.5%

The performance is very similar to the original model in the main manuscript.

3) The table presented as part of Figure 1 reports confidence intervals for sensitivity, specificity and ROC for RPART, SVM, GBM and RF models. Some of those confidence intervals show substantial overlap (e.g., ROC intervals are very similar for SVM, GBM and RF). The authors may want to conduct a statistical test on the classifier performance metrics to validate whether or not their outcomes are significantly different. Several literature references on the topic of model comparison can inform on how to conduct this step (e.g.,).

 According to the reviewer comment, we have reported the difference in ROC statistics and F statistics for the considered MLT method in the figure S2 in the Supplementary Material, specifying the methods used in the methods section.

For each metric, pairwise differences were computed and tested to assess if the difference was equal to zero (3,4). The Bonferroni correction was used to calculate p values and adjust the confidence interval limits for the reported differences.

Considering the ROC Metric, the RF outperforms all the considered algorithms; the RPART algorithm instead represents the worst-performing algorithm in comparison with the other considered algorithm (Table S2, panel A).

From the point of view of the F-statistic the RF performs better than SVM and GBM (Table S2, panel B).

Round 2

Reviewer 2 Report

The authors have responded with a literature citation that class imbalance is not an issue when the minority class makes 22% of the total sample. They also reported a ROC curve to support their point. However, they should still make sure that the models perform substantially better than the null (namely, predicting the majority class all the time), and also that the confusion matrices look reasonable.

Author Response

The authors have responded with a literature citation that class imbalance is not an issue when the minority class makes 22% of the total sample. They also reported a ROC curve to support their point. However, they should still make sure that the models perform substantially better than the null (namely, predicting the majority class all the time), and also that the confusion matrices look reasonable.

We provide further details according to the reviewer’s comments. The supplemental analyses have been provided as supplementary material.

The hypothesis of difference between the estimated model and the null model (which assigns the final prediction to the class with the highest frequency) was tested, reporting the values of roc, sensitivity, and specificity, within the resampling procedures, for the null model and for the model with the best performance, i.e., the random forest model. The difference between the performances was then tested using the method of Hothorn et al. (1) and Eugster et al. (2). Significant differences between the null model and the RF tool (P<0.001) have been identified for sensitivities, specificities, and ROC.

McNemar's test P-value has been also computed (p=0.63) suggesting that the accuracy of the model is not higher than the Non-Informative Rate (the largest proportion of the observed classes).

The average confusion matrix within resampling has been also reported.

         Reference

  Prediction Alive Dead

      Alive   258   34

       Dead   39    44

    Balanced Accuracy: 0.72  

The RF model performs better in correctly classifying the majority class; however, the Balanced Accuracy (0.72), accounting for class imbalance issue, is deemed a good model performance (3).

  1. Hothorn T, Leisch F, Zeileis A, Hornik K. The design and analysis of benchmark experiments. Journal of Computational and Graphical Statistics. 2005;14(3):675–99.
  2. Eugster MJ, Leisch F. Exploratory analysis of benchmark experiments an interactive approach. Computational Statistics. 2011;26(4):699–710.
  3. Bekkar M, Djemaa HK, Alitouche TA. Evaluation measures for models assessment over imbalanced data sets. J Inf Eng Appl. 2013;3(10).
